# Challenging the Conventional Interpretation of HCMV Seronegativity

**DOI:** 10.3390/microorganisms9112382

**Published:** 2021-11-18

**Authors:** Shelley Waters, Silvia Lee, Ashley Irish, Patricia Price

**Affiliations:** 1Curtin Health Innovation Research Institute, School of Medicine, Curtin University, Bentley 6102, Australia; shelley.waters@postgrad.curtin.edu.au (S.W.); silvia.lee@curtin.edu.au (S.L.); 2Department of Microbiology, Pathwest Laboratory Medicine, Fiona Stanley Hospital, Murdoch 6150, Australia; 3Renal Unit, Fiona Stanley Hospital, Murdoch 6150, Australia; Ashley.Irish@health.wa.gov.au

**Keywords:** human cytomegalovirus, seronegative, NK cells, viral IL-10

## Abstract

The majority of adults in the world (around 83%) carry antibodies reactive with HCMV and are thought to retain inactive or latent infections lifelong. The virus is transmitted via saliva, so infection events are likely to be common. Indeed, it is hard to imagine a life without exposure to HCMV. From 45 seronegative individuals (13 renal transplant recipients, 32 healthy adults), we present seven cases who had detectable HCMV DNA in their blood and/or saliva, or a CMV-encoded homologue of IL-10 (vIL-10) in their plasma. One case displayed NK cells characteristic of CMV infection before her HCMV DNA became undetectable. In other cases, the infection may persist with seroconversion blocked by vIL-10. Future research should seek mechanisms that can prevent an individual from seroconverting despite a persistent HCMV infection, as HCMV vaccines may not work well in such people.

## 1. Introduction

It is often assumed that a person who has antibodies reactive against a specific virus (i.e., is defined as “seropositive”) has been infected and may retain that virus. Viruses such as herpesviruses are known to persist for life while other viruses such as influenza are cleared after infection. The corollary is that a person who is seronegative has never been infected. From this, it is a small step to assume that such people have never been exposed to an infectious dose of the virus. Whilst this is logical with novel viruses (e.g., Zika or SARS-CoV-2), it does not fit the clinical data with respect to human cytomegalovirus (HCMV). The majority of adults in the world (around 83%) carry antibodies reactive with HCMV [1]. Active infections are usually controlled in healthy individuals, but the virus enters a latent or inactive state and persists with periodic reactivations. In individuals with acquired immune deficiencies, HCMV infections induce diverse but well-characterized clinical syndromes, and reactivations are relatively common [2,3].

HCMV can be transmitted vertically (transplacentally and through breast feeding) and via organ transplantation and blood transfusions [4]. However, saliva may be the most common route of transmission, as HCMV and murine (M) CMV persist in the salivary gland [5,6]. We have linked the presence of HCMV DNA in saliva with activation of systemic immune responses consistent with a systemic infection [7]. HCMV may pass to a new host through an oral route [4,8], though murine studies suggest the intranasal route effectively initiates primary infection [9]. Transmission via saliva is readily demonstrated in nurseries and crèches [10] but is likely in other situations where individuals are close together. This is not restricted to particular populations.

It is accepted that HCMV establishes latency in fixed tissues such that active replication and viral progeny become undetectable. However, in vitro studies have shown that numerous viral genes continue to be expressed, including UL111a, which encodes HCMV viral IL-10 (vIL-10). Poole and Sinclair [11] dubbed HCMV latency as “sleepless latency”. Throughout life, HCMV undergoes reactivation events and can be shed from asymptomatic hosts or may cause disease.

HCMV vIL-10 is differentially spliced, creating several variants [12]. The best characterized transcripts are cmvIL-10, which is only expressed during lytic infection, and LAcmvIL-10, which continues to be expressed during latency. There is evidence that both variants play a role in immunosuppression with LAcmvIL-10 having more restricted functions. A function of LAcmvIL-10 is downregulation of MHC-II, aiding in the prevention of antigen presentation during latency [13].

Despite the plethora of opportunities to become infected, some people remain seronegative all their lives. We considered whether this represents a failure to seroconvert when infected or rapid clearance of HCMV by specific aspects of the innate immune system. The former is possible as we have identified reports of HCMV DNA in urine from seronegative children aged 4–15 years [14] and in blood from seronegative adults [15]. Furthermore, HCMV pp65- and IE-1-specific CD4^+^ and CD8^+^ T-cells were detected in seronegative renal transplant recipients (RTR) [16]. Here we describe seven seronegative adults who presented evidence of HCMV infections. Our investigations illustrate mechanisms that may contribute to their failure to seroconvert. This includes vIL-10, which interacts with human IL-10R1, initiating signalling via STAT-3. Its biological activity includes the modulation of cellular IL-10 synthesis [13]. HCMV vIL-10 has been detected in plasma [17] with reagents that are available commercially. The reagents utilized in the assay do not cross-react with human IL-10 or EBV vIL-10 and detect both LAcmvIL-10 and cmvIL-10.

## 2. Materials and Methods

### 2.1. Study Cohort

Eighty-two RTR were recruited from renal clinics at Royal Perth Hospital (Western Australia). Inclusion criteria were clinical stability >2 years after transplant, no CMV disease or reactivation within 6 months of sample collection and no current anti-viral treatment. RTR with hepatitis B or C were excluded. Healthy adults recruited as controls were matched with the RTR by age and gender [14]. Ethics approval was obtained from the Royal Perth Hospital Human Research Ethics Committee (EC 2012/155) and endorsed by the Curtin University Human Research Ethics Committee (HR16/2015; HRE2021-0044). The ethics approval complied with the declaration of Helsinki. Participants provided written informed consent. Here we present case studies of five healthy controls and two RTR.

### 2.2. Immunological Responses to HCMV

Peripheral blood mononuclear cells (PBMC) were isolated by Ficoll density centrifugation. Plasma was stored in −80 °C and PBMC in liquid nitrogen. Plasma HCMV IgG titres were assessed using in-house ELISAs based on a lysate of fibroblasts infected with HCMV AD169, recombinant CMV gB (Chiron Diagnostics, Medfield, MA, USA) or IE-1 protein (Miltenyi Biotech, Cologne, Germany). Results were presented as arbitrary units (AU)/mL based on a standard plasma pool [14]. FcRγ^−^ and NKG2C^+^ NK cells (CD3^−^CD56^dim^) and Vδ2^−^ γδ T-cells were enumerated using multicolour flow cytometry, as the populations are expanded in CMV-seropositive RTR [18,19]. Gating strategies are illustrated in Figure 1 and Appendix A. Enzyme-linked immunosorbent spot (ELISpot) assays utilised anti-IFNγ antibodies (MabTech, Stockholm, Sweden). Cells were stimulated with a CMV lysate, CMV pp65 peptide pool (NIH AIDS reagent program, Woburn, MA, USA) or a CMV IE-1 peptide pool (JPT, Berlin, Germany). The number of spots in unstimulated wells were subtracted from the number in stimulated wells and adjusted per 200,000 PBMC.

### 2.3. vIL-10 ELISA

Levels of HCMV vIL-10 were assessed using a sandwich ELISA. First, 96-halfwell plates were coated overnight at 4 °C with 50 μL/well anti-CMV vIL-10 goat polyclonal antibody (AF117, R&D Systems, Minneapolis, MN, USA) at 2 μg/mL. The plates were then washed three times with 150 μL/well of PBS-0.005% TWEEN and blocked for 1 h with 1% BSA/PBS. A standard curve from 4000 pg/mL to 62.5 pg/mL was created using recombinant HCMV vIL-10 with carrier protein (117-VL-025, R&D Systems, Minneapolis, MN, USA). Plasma samples were diluted 1:10 in 1%BSA/PBS and a QC was created by spiking a plasma sample with 500 pg/mL of standard (CV = 17%). A total of 50 uL/well of anti-CMV vIL-10 goat polyclonal biotinylated antibody (BAF117, R&D Systems, Minneapolis, MN, USA) at 0.2 μg/mL was added (2 h, room temperature), followed by a streptavidin-HRP conjugate (30 min, room temperature) and TMB substrate (20 min, room temperature). Reactions were stopped with 25 uL/well 1M H_2_SO_4_ and read at 450 nm. Concentrations of vIL-10 were interpolated from the standard curve [20]. The minimum level detected by this assay was 62.5 pg/mL. The assay was confirmed to specifically detect vIL-10 and not cross react with human IL-10 by replacing the recombinant vIL-10 with recombinant human IL-10 (Human IL-10 DuoSet ELISA, R&D Systems, Minneapolis, MN, USA). No cross-reactivity was observed. Young et al., (2017) confirmed that the reagents used in the vIL-10 ELISA do not cross-react with human IL-10 or EBV vIL-10, and demonstrated detection of LAcmvIL-10 and vIL-10 generated during lytic infections [17].

### 2.4. Detection of CMV DNA in Saliva

Saliva (approximately 5 mL) was collected after a water mouth wash by asking the participant to spit into a tube. Samples were centrifuged (1000× *g*, 10 min). DNA was extracted from the pellet using FavorPrep Blood Genomic DNA Extraction Mini Kits (Favorgen, Ping-Tung, Taiwan) and stored at −80 °C. Each DNA extraction run included a no-sample control with saliva replaced by PBS. HCMV was detected using an in-house qPCR assay with primers targeting the UL54 gene (CMV DNA polymerase) presented in Table 1 [7]. Quantitation was achieved using a standard curve created using DNA extracted from a lysate of HCMV (AD169)-infected fibroblasts and serially diluted 10-fold. Samples were considered positive if a steady amplification curve was initiated before 38 cycles (the lowest point on the standard curve). Positive results were normalized against the gene encoding beta-2-microgobulin and values were reported in arbitrary units (range: 44–721) [21]. Some samples also underwent amplification of MIE by qPCR using the same protocol as UL54 with the primers described in Table 1. All qPCR runs included at least two no-template controls, with PBS controls from the DNA extractions spread across a series of qPCR runs. No amplicons were detected.

To achieve greater sensitivity, samples were also assessed with a nested PCR to detect UL55 [22] (encoding gB; see Table 1), using a PCR buffer with 35 mM MgCl_2_ for the inner reaction and 30 mM MgCl_2_ for the outer PCR in a total volume of 20 µL, with 1 µL of each primer (10 µM), 2 µL PCR buffer, 1 µL 40 mM dNTPs, 0.2 uL Platinum Taq DNA Polymerase (Invitrogen, Carlsbad, CA, USA) and 5 µL DNA (diluted 1:2) for the outer PCRs, and 3 µL of the outer PCR product for the inner PCRs. The cycling conditions for the outer PCRs were 1 cycle of 5 min at 95 °C followed by 35 cycles of 30 s at 95 °C and 30 s at 60 °C and then 1 cycle of 1 min at 72 °C. The inner PCRs ran only 30 cycles. PCR products from the inner PCRs were run on 1% agarose gel in 0.5× TBE buffer stained with GelGreen (Biotium, Fremont, CA, USA) for 1 h at 120 volts. Amplicons were sent for purification and Sanger sequencing by the Australia Genome Research Facility (AGRF). All nested PCR runs included a no-template control carried from the outer PCR to the inner PCR, a no-template control for the inner PCR and DNA extracted from uninfected fibroblasts. No amplification was detected.

### 2.5. Detection of CMV-Encoded miRNA in Saliva

Saliva pellets were thawed and mixed with TRI reagent (1:4) before RNA extraction using the MagMAX—96 for Microarrays kit, as described previously [23] (Applied Biosystems, Foster City, CA, USA). Custom reverse transcription primer pools were generated, and cDNA synthesis for all miRNA assays was performed in a single reaction, according to the manufacturer’s protocols (Applied Biosystems, PN 4465407). A pre-designed primer and probe assay targeting mature miRUS5-2-3p (assay ID: 469255_mat was purchased from Applied Biosystems). RNA from HCMV seronegative healthy participants and uninfected THP-1 cells were used to ensure specificity. These showed no amplification up to 40 cycles. Sensitivity was determined using 10-fold serial dilutions of HCMV AD169 RNA. Samples with cycle thresholds below 10^4^ dilution of the standard (i.e., after cycle 32–36, depending on the miRNA assayed) were considered negative. All samples were run two to four times and called positive if at least two replicates produced amplification.

## 3. Results

Eighty-two RTR and eighty-one healthy controls were recruited in 2014 and tested for HCMV-reactive IgG in plasma, HCMV UL54 DNA in saliva and vIL-10 in plasma and saliva. There were no differences in gender or ethnicity between the two groups, but the RTR were marginally older (54 (21–86) vs. 57 (31–76); *p* = 0.07; Mann–Whitney test, data available in [24]). Levels of CMV reactive antibodies were determined using in-house ELISAs, recognizing a lysate of fibroblasts infected with HCMV AD169 (“HCMV lysate”), gB or IE-1 protein. The cut-off defining seropositivity was 3600 AU/mL based on the HCMV lysate [25,26]. This cut off was determined in relation to samples from individuals deemed to be seronegative using the ARCHITECT CMV IgG assay (Abbott Diagnostic Systems, Lake Forrest, IL, USA). Determinations of HCMV serostatus of RTR were concordant with the ARCHITECT CMV IgG assay according to clinical records. Determinations of serostatus were identical when based on gB or IE-1 (data not shown).

Using this cut-off, 13/82 RTR and 32/81 healthy controls were seronegative (χ^2^, *p* = 0.0007). Here we present seven individuals who were HCMV seronegative by all three ELISAs. Additionally, all seven cases were had no T-cell responses to CMV lysate, IE-1 and pp65 according the ELiSpot assay. Cases 2–7 returned for follow-up in 2017. All remained seronegative by the same ELISA assays, and none had detectable HCMV DNA in their saliva as assessed by in-house UL54 qPCR assay. T-cells were not re-assessed.

Cases 1, 2 and 3 had detectable HCMV DNA in their saliva samples, whilst Cases 4, 5, 6 and 7 had detectable HCMV vIL-10 in their plasma (Table 2). No HCMV vIL-10 was detected in saliva samples from seronegative or seropositive individuals, but vIL-10 was detectable in plasma from 3/32 seronegative healthy controls and 0/13 seronegative RTR in 2014 with one seronegative RTR having detectable vIL-10 in 2012 and 2017 (not 2014). In 2017, 4/23 seronegative healthy controls had detectable vIL-10, including the three who were positive in 2014.

### 3.1. Three Individuals Were Seronegative despite Detectable HCMV DNA

Case 1 is a 33-year-old female RTR who received a donor kidney six years before sample collection. She had remained free of graft rejection, diabetes, HCMV or cardiac disease since transplantation and was stable on tacrolimus, mycophenolate mofetil and prednisolone. Her transplanted kidney was from a HCMV-seronegative donor. Accordingly, there was no detectable HCMV IgG, IgA and IgM in her saliva or plasma (Table 2 and unpublished data). She had no HCMV-reactive T-cells when assessed by EliSpot assay. Case 1 was HCMV DNA positive in saliva by our in-house qPCR assays, detecting MIE and UL54, but negative in plasma and buffy coat. Sanger sequencing of amplicons encompassing gB produced by nested PCRs of the saliva identified a mixed infection where the predominate genotype was gB2. The presence of multiple strains was confirmed by sequencing the genes encoding the NK cell receptor homologues UL18 and UL40 (data not shown). HCMV vIL-10 was not detectable in plasma in 2014 or 2017.

Case 2 is a 57-year-old female healthy control who was seronegative for HCMV IgG in plasma and HCMV DNA negative by qPCR assays targeting MIE and UL54 in plasma and the associated buffy coat samples. However, she had detectable HCMV DNA in her saliva when assessed with the nested PCR targeting gB (UL55). Sanger sequencing identified a mixed infection where the predominate genotype was gB2. HCMV-miR-US5-2-3p was also detected in her saliva [23]. Neither IgA, IgG and IgM antibodies reactive with HCMV antigens nor HCMV vIL-10 were detectable in her saliva or plasma (Table 2; data not shown). She had no HCMV-reactive T-cells when assessed by EliSpot assay. Analysis of her PBMC by flow cytometry revealed expanded populations of FcRγ^−^ and NKG2C^+^ NK-cells, characteristic of HCMV infection [19,27] (Table 2). The FcRγ^−^ population comprised 27.3% of CD3^−^CD56^dim^ NK-cells, compared with a median (range) of 9.1% (5.4–19.1) for other seronegative healthy controls (Figure 1). The NKG2C^+^ population comprised 7.1% of CD3^−^CD56^dim^ NK-cells, compared with a median (range) of 2.6% (0.9–4.0) for other seronegative healthy controls (Figure 1). The same phenotypes were assessed at 2017 showing populations with 15.5% FcRγ^−^ and 11.1% NKG2C^+^ CD3^−^CD56^dim^ cells. In 2017, she had not seroconverted and no HCMV DNA was detected in her saliva. This case suggests the possibility that NK cells can control HCMV replication without seroconversion.

Case 3 was also a healthy female with detectable HCMV DNA in her saliva. She was seronegative in 2014 and 2017, and had no T-cell responses when assessed against three HCMV antigens (Table 2). HCMV vIL-10 was not detectable in her plasma in 2014 or 2017, and no expanded populations of FcRγ^−^ and NKG2C^+^ NK-cells were evident. Expanded populations of Vδ2^−^ γδ T-cells are associated with HCMV seropositivity in RTR and healthy controls [18]. Case 3 had a detectable population of Vδ2^−^ γδ T-cells (2.2% of all γδ T-cells; Table 2), compared with a median (range) of 0.47% (0.06–2.08) for other seronegative healthy controls and 0.96% (0.05–3.8) for seropositive healthy controls.

### 3.2. Four Seronegative Individuals Had Detectable HCMV vIL-10 in Plasma

Case 4 is a 55-year-old male healthy control who was seronegative for HCMV IgG in plasma by our in-house ELISA assays and HCMV DNA negative by our in-house qPCR assay targeting UL54 in saliva. He works in a hospital environment and is frequently exposed to patients with active HCMV infections, but remained seronegative when tested in 2012, 2014 and 2017. However, HCMV vIL-10 was detected in his plasma at all three time points (1840, 639 and 916 pg/mL) vs. the median (range): 166 (0–1440) for seronegative healthy controls tested in 2014. The case establishes the possibility that vIL-10 may suppress seroconversion. This is supported by Cases 5, 6 and 7 (Table 2). Case 5 was an RTR who failed to seroconvert when given a HCMV-positive donor kidney. Case 6 remained seronegative with detectable plasma vIL-10 from 2006 to 2021. Cases 4–7 had no HCMV-reactive T-cells when assessed by EliSpot assay (Table 2).

## 4. Discussion

HCMV seronegativity is widely assumed to define an individual who is not carrying the virus and has not done so recently. Some seronegative transplant recipients face complications following organ transplantation despite prophylaxis and receiving an organ from a HCMV seronegative donor [28].

From a cohort of 45 (32 controls and 13 RTR) individuals we have identified seven (5 controls and 2 RTR) individuals who remained HCMV seronegative despite evidence of current HCMV replication or latent carriage—specifically HCMV DNA, miRNA, vIL-10 and/or FcRγ^−^ NK cells or Vδ2^−^ γδ T-cells characteristic of HCMV seropositivity. We consider the possibility that these cases can be explained by low viral load and poor persistence of viral replication. However, many health adults retain readily detectable HCMV-reactive antibodies and T-cells throughout their lives in the absence of detectable HCMV DNA using highly sensitive PCR assays. It could be argued that these seven individuals mount antibody responses selectively to proteins in the UL133–UL154 region that is deleted in AD169, or UL128–131, which are rendered non-functional by to a frameshift mutation in UL131 in AD169 [29,30]. However, we know of no mechanisms that would explain this selectivity.

Cases 1, 2 and 3 had HCMV DNA in saliva, and Case 2 also had detectable HCMV-miR-US5-2-3p [23]. Case 2 had a striking population of FcRγ^−^ and NKG2C^+^ NK cells in circulation. Several studies have linked this population with HCMV seropositivity [27]. We have linked detection of UL54 HCMV DNA in saliva from the same cohort of RTR with systemic markers of HCMV infection [7]. Here we show that Case 2 cleared the salivary infection without seroconversion, since she was DNA and antibody negative in 2017. Her NK cell response may have favoured compartmentalization to the saliva. This may also be achieved through intrinsic immunity mediated by restriction factors (RS), such as interferon gamma-inducible protein 16 (IFI16), which can prevent HCMV DNA sensing by inhibiting UL54 (DNA polymerase) by interacting with CMV pp65 [31]. IFI16 has been found in saliva [32]. Case 3 had a detectable population of Vδ2^−^γδ T-cells, which may have cleared the virus sufficiently to prevent seroconversion [33] despite the HCMV DNA found in her saliva.

Cases 4–7 had measurable levels of HCMV-encoded vIL-10 in their plasma, which were relatively stable over several years. No individuals had detectable HCMV-reactive T-cells in 2014. Case 6 had detectable vIL-10 over a 15-year period (2006–2021). All samples were tested for HCMV-reactive antibodies and remained negative. Young et al. (2017) detected vIL-10 in single samples from a few seronegative donors [17]. We show that levels of vIL-10 can be relatively stable in seronegative individuals over time.

The detection of vIL-10 may reflect transcription of the encoding gene (UL111a) during HCMV latency [12], explaining the lack of detectable HCMV DNA. We postulate that these individuals harbor latent or replicating HCMV in tissue(s) other than blood or the salivary gland.

We considered the possibility that the presence or absence of detectable vIL-10 reflects variations in UL111a [23]. Viruses identified in Cases 1, 2 and 3 were subjected to amplicon-based enrichment for next generation sequencing of a selection of HCMV genes, including UL111a. The sequence from all three cases had variations when compared to the Toledo reference strain but none were unique to these three samples; i.e., they were also seen in viruses sequenced from HCMV seropositive HIV^+^ individuals from Jakarta (Indonesia; [24]) and RTR and/or healthy controls from Perth, Australia (manuscript in preparation).

The detection of vIL-10 was not rare in the seronegative samples tested (4/32 healthy adults and 1/13 RTR tested between 2014–2017). Case 6 had antibodies reactive with Epstein–Barr virus (EBV) and all cases are healthy adults active in the community, so any immunosuppressive effects of vIL-10 are presumably HCMV-specific. The individuals also had no T-cell responses to three HCMV antigens (Table 2), so vIL-10 may also reduce specific T-cell responses. However, *proof* that vIL-10 downregulates immune responses to HCMV awaits more extensive studies, including further clinical surveys (with analyses of responses to other vaccines) and animal model studies to establish causality. The latter will be complicated as human and mouse CMV show poor homology. Biological effects of vIL-10 to be investigated include enhanced signalling of the CXCR4/CXCL12 pathway triggered by the chemokine receptor homologue US27 [34]. This is distinct from the Stat3-dependent effects of vIL-10, which include modulation of dendritic cell function and macrophage activation [35].

Blockage of HCMV seroconversion by vIL-10 has implications for the efficacy of HCMV vaccines. HCMV vaccines currently in phase I or II clinical trials seek humoral responses to selected HCMV antigens, but their efficacy remains low (i.e., <50% [36]). Indeed, a blockade of vIL-10 has been proposed as a vaccine strategy [35]. Our data supports further development of this approach.

## 5. Conclusions

We have shown that a proportion of HCMV seronegative adults present evidence of active or latent HCMV infections. These may be transient with control by NK cells (as in Case 2) or persistent with seroconversion blocked by vIL-10. It is clear that seropositivity is a risk factor for adverse outcomes following renal transplantation [37]. However, some seronegative individuals still face complications following organ transplantation and HCMV may play a role.

## Figures and Tables

**Figure 1 microorganisms-09-02382-f001:**
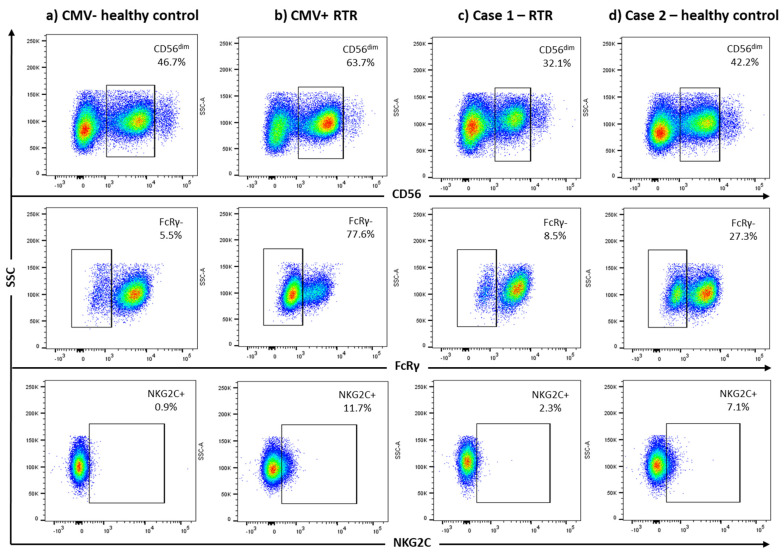
**FcRγ and NKG2C expression on NK cells.** Representative flow cytometry plots for a CMV seronegative healthy control (**a**) and a CMV seropositive renal transplant recipient (RTR) (**b**), Case 1 (**c**) and Case 2 (**d**). Gating strategy: singlets were first defined by forward scatter area (FSC-A) and forward scatter height; lymphocytes were then gated based on the side scatter area and FSC-A, and dead cells were excluded based on uptake of Fixable Viability Stain. Lack of FcRγ expression (middle panel) and expression of NKG2C (bottom panel) were assessed in NK cells identified as CD3^−^CD56^dim^ (top panel). Antibodies used were BUV395-anti-CD3 (clone UCHT1), PE.Cy7-anti-CD56 (clone MEM-188; Biolegend, San Diego, CA, USA) and APC-anti-NKG2C (clone 134591; R&D systems, Minneapolis, MN, USA), and FITC-anti-FcRγ (Merck Millipore, Darmstadt, Germany).

**Table 1 microorganisms-09-02382-t001:** Primer and probe sequences used to detect HCMV DNA by qPCR or nested PCR.

Target	Primer	Sequence (5′-3′)	Product (Base Pairs)

	FWD	CCCGAAAACGTGTCGCC	
UL54	REV	AAACGTTGACGCAGATACTGTAGC	105
	PROBE (5 µM)	6-FAM-TATCGTCAGCATCTGGTGC-BHQ-1	
	FWD	AACTCAGCCTTCCCTAAGACCA	
MIE	REV	GGGAGCACTGAGGCAAGTTC	76
	PROBE (2 µM)	6-FAM-CAATGGCTGCAGTCAGGCCATGG-TAMRA-6	
	FWD	TGAGTATGCCTGCCGTGTGA	105
B2M	REV	ACTCATACACAACTTTCAGCAGCTTAC	
	PROBE (5 µM)	6-FAM-CCATGTGACTTTGTCACAGCCCAAGATAGTT-TAMRA-6	
gB (UL55)	OUTER FWD	GAATRGCTGAYGGRTTGATCTTG	590
	OUTER REV	GATCTCCTGGGATATACAGGACG	
	INNER FWD	GAGTTCCTTGAAGACCTCTAG	
	INNER REV	ACYTTCTGGGAAGCCTCGGAACG	519

**Table 2 microorganisms-09-02382-t002:** Characteristics of seven HCMV seronegative individuals with evidence of HCMV infection.

	Case 1	Case 2	Case 3	Case 4	Case 5	Case 6	Case 7
Age (years)	33	57	62	55	56	34	42
Male (M)/Female (F)	F	F	F	M	M	F	F
Ethnicity	Caucasian	Caucasian	Caucasian	Caucasian	Caucasian	Asian	Caucasian
RTR/Healthy	RTR	Healthy	Healthy	Healthy	RTR	Healthy	Healthy
Donor HCMV status	Negative	-	-	-	Positive	-	-
** *T-cell responses (EliSpot assay presented as cells producing interferon-* ** **γ*/200,000 PBMC) ^a^***							
HCMV lysate	0	0	1	0	3	0	3
IE-1 pooled peptides	1	0	1	0	0	0	1
pp65 pooled peptides	0	0	1	0	1	0	2
** *NK cells (flow cytometry)* **							
FcRγ^−^ (% CD3^−^CD56^dim^)	8.5%	27.3%	2.7%	3.9%	8.8%	4.1%	13.8%
NKG2C^+^ (% CD3^−^CD56^dim^)	2.3%	7.1%	4.3%	1.8%	2.1%	2.8%	2.8%
** *γδ T-cells (flow cytometry)* **							
Vδ2^−^ (% CD3^+^)	0.19%	0.5%	2.2%	0.1%	0.4%	0.7%	1.2%
** *HCMV DNA/miRNA in saliva* **							
HCMV DNA (UL54 qPCR)	Pos	Neg	Pos	Neg	Neg	Neg	Neg
HCMV DNA (UL55 nested PCR)	Pos	Pos	Pos	Neg	Neg	Neg	Neg
HCMV-encoded miRNA	Neg	miR-US5-2-3p	Neg	NT	NT	Neg	NT
** *Plasma HCMV vIL-10 (pg/mL)* **							
2006	NA	NA	NA	NA	NA	801	NA
2012	<60	NA	NA	1840	945	NA	NA
2014	<60	<60	<60	639	<60	1440	166
2017	NA	<60	<60	916	704	1247	786
2021	NA	NA	NA	NA	NA	815	NA

NT: not tested; NA: no sample available. ^a^ Median (range) values for T-cell responses to CMV lysate in seropositive individuals are 245 (0-2077) EliSpots/200,000 PBMC.

## Data Availability

All data described in this paper are available on request.

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
