# Peer review of "Challenging the Conventional Interpretation of HCMV Seronegativity"

_microorganisms, 2021, doi:10.3390/microorganisms9112382_

Round 1

Reviewer 1 Report

To this reviewer, the data presented still tell an incomplete story.

The authors state in their rebuttal that “Assessments of seropositivity, T-cell responses, HCMV DNA and vIL-10 were made systematically in 2014 and 2017 (See Table 1)” (though I think they mean Table 2). So, were serostatuses, T cell responses and DNA PCRs carried out in both 2014 and 2017 for all patients (I know this is stated for e.g. patient 3 in the text) but it is not consistently stated for the other cases and not clear in Table 2. These data are important - for instance, what was the level of detection of HCMV by PCR in patient 5 in 2014 (when vIL-10 levels were very low)?

Prevention of HCMV seroconversion by vIL-10 has enormous implications and that is why the authors really need to obtain a number of samples (not just one) from donor positive/recipient negative transplant patients and show statistically that vIL-10 expression correlates with a lack of seroconversion and PCR negativity in these individuals.

Minor comments: PCR primers are shown in Table 1 not Table 2 as stated in “Detection of CMV DNA in Saliva”. And reference to Table 1 when the authors mean Table 2 is made in multiple places in the text.

Author Response

See attached response

Reviewer 2 Report

In this revised manuscript the authors have addressed some of the points raised in my review and addressed other points in their response to my review. 

I am mostly happy with the response and manuscript revisions, however, some of the sections of revised text (shown in blue in the manuscript) require proof reading as I noted some typographical errors. 

Additionally, the new section on Latency in the introduction could be improved, particularly as vIL-10, a transcript shown to be expressed during latency, is a key part of the findings in this manuscript.

Within the discussion, the nuances of differences in serological testing and functional testing could be discussed still.  Consideration of whether neutralising capacity of CMV antibodies may be present in the absence of detectable IgG responses, the role of antibodies to the pentameric complex (which may be missed when using AD169 lysate due to the large sections of the CMV genome lost in this lab adapted strain) and whether there are differences in avidity of the antibody response in these 7 cases.  While I agree that there are definite cases where CMV specific antibodies are not detectable, for the sake of completeness all possible explanations for this should be explored in the discussion.

Author Response

See attached response

Round 2

Reviewer 1 Report

For this reviewer, a key take home message of this paper is still the suggestion that vIL-10 expression inhibits host immune responses to HCMV. The authors state on line 301 “The detection of vIL-10 was common in the seronegative samples tested (4/32 healthy adults and 1/13 RTR tested between 2014-2017).” These numbers hardly argue that detection of vIL-10 was common in the seronegative samples. For this reviewer, there are still too many suggestions/suppositions in the manuscript which are not experimentally interrogated. However, I accept that the authors have tried to underscore this in some of their revisions.

Author Response

We are not certain what changes you wanted but have clarified critical issues in the Discussion. These changes are in red in the manuscript.

This manuscript is a resubmission of an earlier submission. The following is a list of the peer review reports and author responses from that submission.

Round 1

Reviewer 1 Report

This is an interesting paper that extends previous work by others suggesting that CMV infection might occur without development of IgG responses. While I happen to agree with the authors that CMV infection can occur without humoral responses, there are several issues that limit enthusiasm for the current report that are outlined below.

Major Concerns:

  1. Levels of CMV reactive antibodies were determined using in-house ELISAs recognizing a lysate of fibroblasts infected with HCMV AD169 (“HCMV lysate”), gB or IE-1 protein – Is this possibly an assay sensitivity/specificity issue? Alternatively some people might not make antibody to these particular proteins – it is clear from T-cell data that not all people make T-cells to the same proteins. This merits discussion. If this is an antibody detection issue, it is possible that these folks have antibodies that aren’t being tested for and detected.
  2. There is convincing evidence of infection without seroconversion here in 3 patients – specifically those with CMV recovered from their saliva. There is pretty convincing evidence that two more have seronegative CMV – those with CMV-specific T-cells on elispot. The other two are circumstantial given the vIL-10 results, and IMO should be identified as possibly infected and seronegative. The NK and delta gamma T-cell data are wholly circumstantial and in my mind do not strongly support the supposition of seronegative but virus positive, but do not contradict them either.
  3. The biggest quandary here are the vIL-10 results. To have highly detectable vIL-10 levels and not have any other viral gene detectable is surprising. This implies latent viral transcription of vIL-10 and in my mind to give such levels of vIL-10 there must be a significant pool of latently infected cells somewhere. This merits discussion IMO.
  4. The authors postulate “The detection of vIL-10 may reflect transcription of the encoding gene (UL111a) during HCMV latency [22], explaining the lack of detectable HCMV DNA.”. This is testable. Latency associated vIL-10 is a splice variant that should be detectable in these patients given their high vIL-10 levels that should be evaluated to distinguish latent transcription versus active infection.
  5. There were also seronegative patients with positive vIL-10 reported in reference 13 (Young et al) that should be cited here also -ie the finding that some “seronegative” patients can have vIL-10 expression is not novel.
  6. Near the end of the discussion, there is a fair amount of hand-waving over the immune impact of vIL-10. This reviewer considers IL-10 to be both immune suppressive and stimulatory to B-cells, which is inconsistent with the hypothesis that vIL-10 impedes a humoral response. Shouldn’t an immune suppressive agent also enhance infection and B-cell responses? This makes the latency associated vIL-10 question posed above even more relevant. It has also been suggested by others that the vIL-10 and LATvIL-10 might behave differently in-vivo than cIL-10, but this is speculative here at best.
  7. Meanwhile the possibility that all of this is viral load related is ignored. There are data showing that naturally infected hosts (human, pigs, mice) have variable viral loads in their tissues (Reviewed by Marandu MMI 2019). There are clear data from murine models that antiviral responses correlate with host viral load at the time of primary infection (see work from Redeker 2014 and Trgovcich 2016). There are also literature showing that very low level infections can occur without development of CMV-specific T or B cell responses (Thomas et al 2010 Transplant Immunology).
  8. Last in the discussion is a paragraph on vaccination. The data presented in the current report are insufficient to make the argument to inhibit vIL-10 as a vaccine strategy.
  9. Somewhere in the manuscript the authors need to clarify the cross-reactivity (or lack thereof rather) between cIL-10, vIL-10, and other viral IL-10 such as EBV etc. Particularly important for patient 6 that is known EBV positive and HCMV seronegative…is the EBV vIL-10 cross reacting?
  10. FcγR- NK cells or Vδ2- γδ T-cells characteristic of HCMV seropositivity – this is circumstantial at best weak evidence of infection. I think that this should be acknowledged as supporting evidence and not prima facie evidence of infection.

Minor concerns

  1. Page 1 “murine studies suggests” should be “murine studies suggest”
  2. Pg 5 Discussion – “No individuals had no detectable HCMV-reactive T-cells in 2014.” – please change this double negative sentence
  3. Conclusion: ….“However some 187 seronegative individuals still face complications following organ transplantation and HCMV may play a role.” Never mentioned anywhere in the manuscript….deserves discussion before the last sentence.

Reviewer 2 Report

The manuscript by Waters et al reports that not all individuals apparently HCMV seronegative individuals are also PCR negative for viral genome or markers of having seen virus. They also suggest that expression of viral IL-10 may suppress seroconversion to HCMV positive status.

The first thing to be said is that it is already well established that HCMV seronegative individual have been shown to apparently carry HCMV genome in their peripheral blood as detected by sensitive (often nested) PCRs. The authors do allude to this in their reference to work by Larsson et al (1998) but this observation has been made by others previously (Stanier et al 1989; Taylor-Wiedemann et al 1991) and with similar frequencies of PCR positive/seronegative donors as described in the present manuscript. It is also accepted that the clinical relevance of this is unclear as HCMV donor negative/recipient negative scenarios rarely if ever result in problems with HCMV reactivation.

So essentially, the primary observations described by the authors are not novel.

The authors go on to show that, using single patient studies, association of some markers of apparent HCMV infection with lack of seroconversion to HCMV seropositivity. Unfortunately, the data presented are piecemeal and have internal inconsistencies with respect to consistent markers of HCMV infection (or analyses of those markers were not carried out due to lack of sample). The view that expression of vIL-10 in case 4 suppressed seroconversion is interesting but a single observation - although I accept that some cases (e.g. case 5) showed a roughly similar phenomenon with respect to vIL-10 expression and lack of seropositivity.

In order to support the main "take home" message of the manuscript (on the basis that the occurrence of HCMV PCR positive/seronegative individuals is well established and not a novel observation), the authors really need to e.g. obtain a number (and this could be small) of samples from donor positive/recipient negative transplant patients and show statistically a lack of seroconversion in these individuals correlated to PCR negativity but e.g. vIL-10 expression.

Reviewer 3 Report

In this interesting and relevant study the author investigate evidence of CMV infection in seronegative individuals (healthy adults and renal transplant recipients). The authors convincingly show the presence of HCMV DNA in saliva of a subgroup of seronegative individuals by using a stringent nested PCR for UL55 viral gene and confirm by HCMV miRNA in a smaller subset. My only concern is that the author might have missed some of the seropositives by only running ELISA for IgG. They do state in line 96: Neither IgA, IgG and IgM antibodies reactive with HCMV 96 antigens nor HCMV vIL-10 were detectable in saliva or plasma (Table 1; data not shown). I did not find that data in Table 1 and would prefer if it’s shown as showing that these individuals are also IgM negative would exclude the remote possibility of a very recent infection. Other than that study is well designed, sample size is adequate and research methods are appropriate.

Reviewer 4 Report

Waters and colleagues present interesting data from a small cohort of CMV seronegative donors who have other evidence of CMV infection in the absence of detectable CMV specific antibodies.  The authors examined 45 seronegative individuals in total and found seven cases where either CMV DNA was detected in saliva or CMV vIL-10 was detected in plasma samples from these donors. 

Whilst the data is of interest and should be reported, in its current form the manuscript fails to convey the information with sufficient impact.  Improvements to the presentation of the data from the seven subjects described are necessary as well as revision of both the introduction and discussion sections of the manuscript.

Specifically, I noted the following issues with the introduction:

a) The opening sentence (lines 22 – 23) is factually incorrect as certain viruses e.g., influenza are cleared and not retained by an individual. This statement on really applies to viruses e.g., members of the herpes virus family which are known to persist.

b) The final sentence of the opening paragraph (lines 28 – 32) requires a reference.

c) Lines 34 – 36, the cited references do not support the statement that CMV establishes latency in the salivary gland. The reference by Amin et al establishes the important role of saliva as a route of transmission for CMV, however the cited murine reference by Thom et al describes antigen presenting cells (APCs) present in the salivary gland (SG) and again does not discuss latency.

d) A brief overview of latent CMV infection in humans would be helpful in this introduction, there are numerous reviews on this subject in addition to primary research papers which could be cited in support of this.

e) A minor quibble, but the language used in line 44 referring to the avoidance of infection with CMV as a “lifelong failure” seems inappropriate.

Results section:

f) Line 60 a p-value is cited with no back up data or description of statistical test used – as such it is meaningless.

g) Were the seven discordant seronegative donors serology tested elsewhere by any other commercial or approved serological methods? This is an interesting point which should also be addressed in the discussion – as variations in serology results between different tests have been observed by many others and is referred to in the comments in the following WHO report, Wissel N et al. Report of the WHO Collaborative Study to establish the First International Standard for Detection of IgG antibodies to Cytomegalovirus (anti-CMV IgG) WHO/BS/2017.2322 – Expert Committee on Biological Standardization. Geneva: World Health Organization (2017). Specifically comment 8 on page 16 of the report. Furthermore, were the “seronegative” samples tested for avidity or neutralisation capacity? As, this may show the presence of CMV specific antibodies that are not revealed via the IgG ELISA.  If not consideration of these discrepancies in the discussion would be helpful.

h) Whilst reported in Table 1 the T cell data is not discussed in the main results section.

i) There is various different uses of FcRgamma in the manuscript (FcRγ and FcγR) also it is not clear but is this actually referring to FcεRIγ chain expression?

j) I feel the data for section 2.2 regarding the detection of HCMV vIL-10 in four donors could be presented more clearly – perhaps as a separate figure rather than in the bottom of Table 1. This is interesting data which is rather lost in the current manuscript.

k) While the NK phenotype data is shown in figure 1 the gamma delta T cell phenotype data is merely described and not shown – could this be added (possibly as a supplementary figure).

l) Line 106 – One case does not really “establish” that NK cells control HCMV replication – a different phrase should be considered.

Discussion:

m) Data regarding Latent viral carriage in monocytes and CD34+ cells in humans has been demonstrated in multiple publications recently including Parry et al 2016 doi:10.1186/s12979-015-0056-6 and Jackson et al 2017 doi: 3389/fimmu.2017.00733. It would be interesting to consider whether in these seven discordant individuals latent CMV could be detected and should be discussed.

n) The absence of serum anti-CMV antibodies does not necessarily mean absence of anti-CMV antibodies from the donor – as suggested in point g) above, consideration of other possibilities to explain the absence of detectable serum antibodies by the three ELISA methods used here should be discussed.

o) The detection of CMV specific T cell responses in the absence of serum antibodies has been published previously and one of these papers Litjens et al 2017 is included in the introduction. In the paragraph hypothesising that the CMV vIL-10 may be inhibiting T cell responses (lines 168 – 177) consideration of the diversity of the CD4 and CD8 T cell response to different CMV proteins as shown by Sylwester et al 2005 doi: 1084/jem.20050882, and that not everyone who has CMV reactive T cells necessarily responds to pp65 or IE1.  Additionally, in line 168 it states that detection of vIL-10 was common in the seronegative subjects – but I calculated that only 11% of the 45 donors had detectable vIL-10 which is not really common.

Materials and Methods:

p) There is omission of the declaration of Helsinki in the Study Cohort section.

q) A brief overview of the antibodies used for the flow cytometry phenotyping would be helpful.

r) Where are the supplier details for the commercially available antibodies for the vIL-10 ELISA.

s) Has the 1st WHO international standard for human cytomegalovirus (HCMV) for nucleic acid amplification (NAT)-based assays been used in the DNA detection assays or as a reference sample for sequencing?

Overall:

t) There are some typographical and spelling errors I noted in the manuscript which should be corrected in the revised manuscript. Including line 11 – comma after indeed; line 76 – spelling of detectable; Table 2 – gB inner primers are both FWD.
